Soil carbon, nitrogen and phosphorus ecological stoichiometry shifts with tree species in subalpine plantations

Qi Kaibin 1 2
Pang Xueyong 1
Yang Bing 1
Bao Weikai baowk@cib.ac.cn 1
1 CAS Key Laboratory of Mountain Ecological Restoration and Bioresource Utilization & Ecological Restoration Biodiversity Conservation Key Laboratory of Sichuan Province, Chengdu Institute of Biology, Chinese Academy of Sciences , Chengdu , China
2 University of Chinese Academy of Sciences , Beijing , China
Livesley Stephen
Electronic publication date: 2020 Oct 12
Publication date: 2020
Volume: 8
Electronic Location ID: e9702
Received 2019 Dec 23; Accepted 2020 Jul 21
Copyright: ©2020 Qi et al.
Copyright year: 2020
Copyright holder: Qi et al.
License: This is an open access article distributed under the terms of the Creative Commons Attribution License, which permits unrestricted use, distribution, reproduction and adaptation in any medium and for any purpose provided that it is properly attributed. For attribution, the original author(s), title, publication source (PeerJ) and either DOI or URL of the article must be cited.
License URL: https://creativecommons.org/licenses/by/4.0/

Keywords: Ecological stoichiometry, Plantation in subalpine region, Soil depth, Litter, Fine root

Funding: Major Science Technology Project of Sichuan Province 2018SZDZXD035 This study was jointly funded by the Major Science Technology Project of Sichuan Province (Grant numbers: 2018SZDZXD035). The funders had no role in study design, data collection and analysis, decision to publish, or preparation of the manuscript.

==============================
Understanding ecological stoichiometric characteristics of soil nutrient elements, such as carbon (C), nitrogen (N) and phosphorus (P) is crucial to guide ecological restoration of plantations in ecologically vulnerable areas, such as alpine and subalpine regions. However, there has been only a few related studies, and thus whether and how different tree species would affect soil C:N:P ecological stoichiometry remains unclear. We compared soil C:N:P ecological stoichiometry of Pinus tabulaeformis, Larix kaempferi and Cercidiphyllum japonicum to primary shrubland in a subalpine region. We observed strong tree-specific and depth-dependent effects on soil C:N:P stoichiometry in subalpine plantations. In general, the C:N, C:P and N:P of topsoil (0–10 cm) are higher than subsoil (>10 cm) layer at 0–30 cm depth profiles. The differences in C:N, N:P and C:P at the topsoil across target tree species were significantly linked to standing litter stock, tree biomass/total aboveground biomass and Margalef’s index of plant community, respectively, whereas the observed variations of C:N, N:P and C:P ratio among soil profiles are closely related to differences in soil bulk density, soil moisture, the quantity and quality of aboveground litter inputs as well as underground fine root across plantations examined. Our results highlight that soil nutrients in plantation depend on litter quantity and quality of selected tree species as well as soil physical attributes. Therefore, matching site with trees is crucial to enhance ecological functioning in degraded regions resulting from human activity.

Introduction

Ecological stoichiometry addresses the equilibrium or interactions of the main elements as well as the correlations between elements and ecosystem functioning (Cambardella & Elliott, 1992; Elser, 2000; Gusewell, 2004). Nitrogen (N) and Phosphorus (P) are the most critical nutrients limiting plant growth, and their balance can regulate biological processes in terrestrial ecosystems (Elser et al., 2007; Gusewell, 2004; Reich & Oleksyn, 2004; Vitousek & Howarth, 1991), such as the process of carbon (C) storage (Hessen et al., 2004; Yu et al., 2015). Soil C:N, a sensitive indicator of the C and N reserves and also a soil quality in ecosystems (Tessier & Raynal, 2003), is bound up with the C allocation and N mineralization rates of soil organic material in forest ecosystems. Soil C:P reflects the ability of soil P mineralization, a low soil C:P favors microorganisms to decompose organic matter and desorb nutrients, thus provides higher soil available P content (Tian et al., 2010). Soil N:P can measure the N saturation status and it is used to determine the thresholds for soil nutrient limitation (Tessier & Raynal, 2003). The biogeochemical cycles of C, N, and P in terrestrial ecosystems are strongly interconnected through the biochemical reactions during primary production, respiration and decomposition (Daufresne & Loreau, 2001; Zechmeister-Boltenstern et al., 2015). A deficiency or lower content of N and P leads to a higher C:N and C:P ratios, and their excess triggers C deficiency (Gusewell, Jewell & Edwards, 2005; Tessier & Raynal, 2003).

Since the C:N:P stoichiometry is regarded as an indicator of soil fertility and the excess/limitation conditions of soil nutrients (Bing et al., 2016), studying its response to disturbance resulting from human activities is essential for the enhancing of ecosystem functioning. Although some studies report that soil C:N:P ecological stoichiometric ratios are well-constrained (Cleveland & Liptzin, 2007), others found that soil C:N:P stoichiometric ratios can be affected by climate factors including temperature and precipitation (Callesen et al., 2007; Vesterdal et al., 2008; Zhang, Li & Wang, 2019), topography factors, such as elevation and latitude (Moser et al., 2011; Whitaker et al., 2014; Xu, Thornton & Post, 2013; Zhang, Li & Wang, 2019), soil texture and vegetation types (Cools et al., 2014; Feng, Bao & Pang, 2017; Tian et al., 2010; Vesterdal et al., 2008; Xu, Thornton & Post, 2013). In addition, ecological stoichiometry can also be influenced by ecological restoration approaches, and afforestation can be a major factor affecting soil C:N:P stoichiometric ratio (e.g.,  Davis et al., 2007; Deng, Liu & Shangguan, 2014). Different tree species planted for restoration of afforestation can influence soil properties through multiple pathways. However, in alpine and subalpine regions whether there are differences in soil C:N:P stoichiometric ratio between afforestation and reforestation remains largely unknown. Moreover, tree species differ from one another in the quality and quantity of litter and root exudate (Aponte, Garcia & Maranon, 2012; Paterson et al., 2007) and this in turn can influence soil C and N mineralization mediated by microbial communities (Alberti et al., 2015; Prescott & Grayston, 2013). Finally, the vertical patterns of C, N and P stoichiometry vary with ecosystem type (Feng, Bao & Pang, 2017). Although a few studies have examined soil C:N:P stoichiometric ratios of forest ecosystems in alpine and subalpine region (Bing et al., 2016; Feng & Bao, 2018; Feng, Bao & Pang, 2017; Mueller et al., 2017; Yang et al., 2014; Zhang, Li & Wang, 2019), studying how C:N:P stoichiometric ratios respond to ecological restoration in depth should include more tree species and more sites.

The main objective of this study was to evaluate the effect of commonly used tree species in plantations on soil C:N:P ecological stoichiometry. Since soil stoichiometric ratios vary with the species characteristics of plantations (Lawrence, Fahey & Zedler, 2013; Sardans, Rivas-Ubach & Penuelas, 2012; Vinton & Burke, 1995), we hypothesized that: (1) the soil C:N:P stoichiometry of plantations would be influenced by the tree species; (2) the variations in soil C: N: P stoichiometric ratio across tree species decrease with increase in soil depth because the influence of litter input on soil nutrient decreases with increase in soil layer; (3) the fine root biomass (FRB), standing litter stock (LS), tree biomass/total aboveground biomass (TB/TAB) and Margalef’s index are major influencing factors of soil C:N:P ecological stoichiometry, whereas the effect size depends on soil depth.

Material and Methods

Study area

The study was conducted at the Mao Country Mountain Ecosystem Research Station (31°37′N, 103°54′E), the Chinese Academy of Sciences, which located in Sichuan province, China. The mean monthly temperature of the study area ranges from −0.9 °C in January to 18.6 °C in July, with an annual mean temperature of 9.3 °C. The growing season is from May to September. The mean annual precipitation is approximately 900 mm, about 70% falls during the growing season. The annual accumulated temperature, greater than or equal to 10 °C is 2635.1 °C. The soil at the study site belongs to a Calcic Luvisol according to the IUSS Working Group WRB (2007). The soil texture was silt loam with 15.5% and 15.3% of sand, 62.5% and 63.3% of silt, 21.9% and 21.5% of clay in the 0–10 cm, and 10–20 cm soil depths, respectively (Jiang, Pang & Bao, 2011).

Forest types and management activities

In August 2007, plantations of Pinus tabulaeformis (PT), Larix kaempferi (LAR), Cercidiphyllum japonicum (CJ) from three different sites were selected and native secondary shrublands dominated by Corylus heterophylla and Quercus liaotungensis nearby were chosen as control, resulting in 4 woodlands (12 plots). We chose PT, LAR and CJ because they were commonly used when restoring or replacing native thicket in western Sichuan province, as was also the case with the study area (Pang & Bao, 2011). These plantations were established with terracing in the spring of 1987 on cutovers of primary thicket, which were clear-fallen in the autumn of 1986. They have not been fertilized since the establishment. Prior to establishment, the main soil properties in these plantations were similar to those of the native secondary coppice forest (Pang & Bao, 2011). In August 2018, we sampled these plots again to examine the changes during the past 11 years. The understory species were dominated by native broad-leaved species, including Quercus aliena, Corylus heterophylla, Rosa spp., Spiraea spp., Phlanis umbrosa, Voila spp., Anaphalis sinica, Potentilla spp., without any species being absolutely dominant (Pang & Bao, 2011). The other basic information about the chosen forests was shown in the Table 1.

Table 1 The basic information of plantation stands.

Species identity	Location	Elevation (m)	Aspect	Slope (°)	Canopy density	Height (m)	DBH (cm)	SD (trees⋅ha−1)	
LAR	31°41′23″E, 103°53′42″N	2070	NE	14	0.9	11.69	16.29	1200	
	31°41′23″E, 103°53′43″N	2070	NE	15	0.8	10.18	11.31	2200	
	31°41′22″E, 103°53′42″N	2081	N	21	0.98	11.62	13.66	1000	
PT	31°41′27″E, 103°53′41″N	2066	N	9	0.85	9.62	8.67	5100	
	31°41′26″E, 103°53′42″N	2065	N	6	0.89	11.68	10.89	4000	
	31°41′26″E, 103°53′43″N	2073	N	20	0.96	10.65	8.29	2700	
CJ	31°41′24″E, 103°53′24″N	2056	NW	10	0.9	11.84	12.15	2900	
	31°41′26″E, 103°53′39″N	2068	NW	19	0.93	11.47	11.97	3400	
	31°41′27″E, 103°53′39″N	2020	NW	18	0.89	13.17	11.6	3300	
S	31°41′36″E, 103°53′42″N	1933	N	17	0.94	5.95	3.87	15100	
	31°41′35″E, 103°53′41″N	1948	NW	22	0.9	3.9	2.81	15300	
	31°41′35″E, 103°53′42″N	1953	NW	18	0.86	5.91	3.47	12500	
Notes.

LAR Larix kaempferi

PT Pinus tabulaeformis

CJ Cercidiphyllum japonicum

S Shrubland

Height Tree height

DBH Diameter at breast height

SD Stand density

Vegetation measurements, soil sampling and analysis

In August 2007 and 2018, three 10 × 10 m standard plot was randomly set in an area of about 0.5 ha for each woodland. The height (H) and the diameter at breast height (DBH) of trees in each plot were measured and stand density were calculated.

In each plot, the aboveground biomass of each layer of understory vegetation (including shrubs and herbs) were recorded using destructive sampling in five 2 m × 2 m quadrates. All aboveground biomass within each sampling category was clipped and dried at 65 °C until the weight was constant. The litter on the soil surface was collected from the same quadrates, mixed, dried at 65 °C and weighted. Soil samples were collected to a depth of 30 cm at three intervals of 0–10, 10–20 and 20–30 cm from 10 random sampling sites along a “W” shape with a soil auger (50 mm diameter). These sampling sites were at least 1.5 m apart from each other and 2 m away from the boundary. The samples from each quadrat were pooled to give one composite sample per plot and depth. The soil samples were taken to the laboratory and soil moisture content were determined with 20 g soil each sample in oven drying method at 105 °C for 24 h. The soil bulk density (BD) was determined using stainless steel cylinders (100 cm3) in triplicate for each plot before soil sampling (Qu et al., 2016). The BD of soil was calculated by dividing the dry mass after oven drying at 105 °C for 24 h of each composite soil sample by the sample volume. The soil samples were air-dried after removing the gravel, soil animals and plant debris and breaking the large fractions. The air-dried soil sample was ground and then passed through 20-mesh (0.9 mm) and 100-mesh (0.15 mm) nylon sieves, respectively(Li et al., 2018). The processed samples were preserved for the determination of soil organic carbon (SOC), total nitrogen (TN) and total phosphorus (TP). SOC and TN were determined by combustion in a Macro Elemental Analyser (vario MACRO, Elementar Co., Germany). The TP was measured using the sulphuric acid-soluble perchlorate acid- molybdenum antimony colorimetric method (Bowman, 1988). C, N, and P contents in leaves, litter, and soil samples were mass-based. The atomic ratios were determined according to the formula: (1) C:N=Ccontent12Ncontent14

(2) N:P=Ncontent14Pcontent31

(3) C:P=Ccontent12Pcontent31

Statistical analysis

Three-factor analysis of variance (ANOVA) followed by Tukey HSD post-hoc analysis was used to determine differences in results for atomic ratios of C:N, C:P and N:P across treatments with target tree species, soil depths and sampling time (2007 and 2018) as factors. Additionally, two-factor analysis of variance (ANOVA) followed by Tukey HSD post-hoc analysis was used to determine differences in results for atomic ratios of C:N, C:P and N:P across treatments in the same soil depths with target tree species and sampling time as factors. Besides, one-factor analysis of variance (ANOVA) followed by Tukey HSD post-hoc analysis and was used to examine the differences in results for atomic ratios of C:N, C:P and N:P across the same sampling times and soil depth between different target tree species. Before analysis, the normality and the homogeneity of the residuals for data were examined by Shapiro–Wilk test and by Kolmogorov–Smirnov test in the ‘stats’ package, respectively. If assumption of ANOVA of a given variable was met, we do ANOVA consequently. Otherwise, the non-parametric Kruskal-Wallis test was performed and the Wilcoxon test was performed in multiple comparisons. For all analyses, the significant level were set at α = 0.05. Besides, the differences between the two sampling events (2018 versus 2007) were compared with student’s t-test or Wilcoxon test. Pearson correlation analysis was used to examine the correlations among TB, understory plant biomass (UPB), TAB, TB/TAB, understory plant biomass/total aboveground biomass (UPB/TAB), FRB, diversity indices of plant community (Richness index, Margale’f index, Shannon-Wiener index, Simpson index and Pielou index), LS, C, N and P content, C:N:P stoichiometric ratios in litter and C:N:P stoichiometric ratios in topsoil. Additionally, the main influencing factors were selected by multiple linear regression using “step-AIC” function (R package: MASS) (Venables & Ripley, 2002) in R version 3.5.2. Furthermore, the corresponding contribution of selected factors were obtained by “relimpo” function (R package: relimpo) (Groemping, 2006) in R version 3.5.2. Finally, the determinant factors of soil C:N:P stoichiometry at the 0–10 cm layers were examined with multiple regression, with FRB, LS, TB/TAB, Margalef’s index, C and P contents in litter as independent variables.

Results

Soil C, N and P stoichiometry

Soil C: N and N:P varied among soil depths and between sampling times (Table 2; Figs. 1A–1F). Soil C:P was responsive to tree species, soil depth, sampling time and interactive effect of tree species by soil depth (Table 2; Figs. 1G–1I).

Table 2 Summary of the linear mixed model showing the effects of soil layer, trees species and sampling time on the C:N, N:P and C:P of soil.

Variables	Depth (D)	Tree species (TS)	Time (T)	D × TS	TS × T	D × TS × T	
	d.f.	F	P	d.f.	F	P	d.f.	F	P	d.f.	F	P	d.f.	F	P	d.f.	F	P	
C:N	2	26.85	<0.001	3	2.04	0.12	1	41.88	<0.001	6	2.91	0.02	3	0.61	0.61	6	2.07	0.07	
N:P	2	53.15	<0.001	3	27.02	<0.001	1	0.00	0.99	6	7.18	<0.001	3	1.01	0.40	6	1.35	0.25	
C:P	2	104.03	<0.001	3	24.27	<0.001	1	23.47	<0.001	6	7.18	<0.001	3	1.90	0.14	6	1.19	0.33	

Figure 1 Soil C:N:P stoichiometric ratio across soil depth, tree species and sampling times.

(A–C) C:N, (D–F) N:P; (G–I) C:P. The capital and lowercase letters indicate significant differences across different tree species (P < 0.05) in 2018 and 2007, respectively. NS, * and ** denote the differences between sampling time (2018 versus 2007) based on t-test or Wilcoxon test are at P > 0.05, 0.01 ≤ P ≤ 0.05 and P < 0.01, respectively.

Dynamics of Soil C, N and P stoichiometry

In 2007, only the C:N ratio of soil at the depth of 10–20 cm varied significantly with tree species (Fig. 1B). Specifically, the highest C:N ratio was observed in soil of the PT plantation, followed by CJ plantation and shrubland, and the lowest C:N ratio was observed in the soil of the LAR plantation (Fig. 1B). In 2018, both the C:N ratios of soil at the depths of 0–10 cm and 10–20 cm varied significantly with tree species (Figs. 1A & 1B). At the depth of 0–10 cm the highest C:N ratio in soil were observed in soil of the LAR plantations, followed by PT and CJ plantations, and the lowest C:N ratio occurred in the soil of the shrubland (Fig. 1A). At the depth of 10–20 cm, the highest C:N ratio in soil was observed in soil of the PT plantation, followed by shrubland and LAR plantations and the lowest C:N ratio was observed in the soil of the CJ plantations (Fig. 1B).

In 2007, the N:P ratio of soil varied significantly with tree species for the depth of 0–10 cm and 20-30 cm (Figs. 1D & 1F). At the depth of 0–10 cm the highest N:P ratio of soil was observed in shrubland, followed by LAR and PT plantations, and the lowest N:P ratio of soil occurred in the CJ plantation (Fig. 1D). At the depth of 20–30 cm, the highest N:P ratio in soil was observed in soil of the CJ plantation, followed by shrubland and LAR plantations and the lowest C:N ratio was observed in the soil of the PT plantations (Fig. 1F). In 2018, only the N:P ratio of soil at the depth of 0–10 cm varied significantly with tree species (Fig. 1D). The trend was same as that at the depth of 0–10 cm in 2007 (Fig. 1D).

In 2007, the C:P of soil at the depth of 0–10 cm varied with tree species (Fig. 1G). Specifically, the highest C:P of soil was observed in e shrubland, followed by LAR and PT plantations, and the lowest C:P of soil was observed in the CJ plantation (Fig. 1G). In 2018, the C:P ratios of soil at the depth of 0–10 cm and 10–20 cm have shown similar trend as 2007, however the soil at the depth of 20–30 cm has shown significant differences among tree species (Figs. 1G–1I). Specifically, at the depth of 20–30 cm the highest C:N ratio of soil were observed in soil of the shrubland, followed by CJ and PT plantations, and the lowest C:N ratio of soil was observed in the LAR plantations (Fig. 1I).

Correlations among soil C:N:P stoichiometric ratios, soil properties and plant community attributes in 2018

Across soil profiles, the C:N, N:P and C:P significantly decreased with the increase in soil bulk density, whereas significantly increased with the increase in soil moisture and FRB (Fig. 2). At the topsoil, the C:N was significantly positively correlated to LS (P < 0.001), whereas negatively correlated to FRB and C content of litter (P < 0.05) (Table 3). The N:P was significantly positively correlated to UPB (P < 0.001), UPB/TAB (P < 0.001), tree & shrub richness (P < 0.001), Margalef’s index (P < 0.001), Shannon-Wiener index (P < 0.001), Pielou evenness index (P < 0.001), but negatively correlated to TB (P < 0.001), TAB (P < 0.001), TB/TAB (P < 0.001), P content of litter (P < 0.05) and Simpson dominance index (P < 0.001) (Table 3). The C:P was significantly positively correlated to tree & shrub richness (P < 0.001), Margalef’s index (P < 0.001), Shannon-Wiener index (P < 0.001), Pielou evenness index (P < 0.001), but negatively correlated to TB (P < 0.001), TAB (P < 0.001), TB/TAB (P < 0.05), C content of litter (P < 0.05), litter C:N (P < 0.05) and Simpson index (P < 0.001) (Table 3).

Soil C:N at the topsoil was affected by LS and FRB (r2 = 0.76, F = 13.96, P = 0.002), and the LS contributed to 67.31% of the variation (Fig. 3). Soil N:P at the topsoil was affected by P content of litter, Margalef’s index and TB/TAB (r2 = 0.98, F = 120.50, P < 0.001), and TB/TAB and Margalef’s index contributed to 48% and 38% of the variation, respectively (Fig. 3). Soil C:P at the topsoil was affected by C content of litter and Margalef’s index (r2 = 0.81, F = 19.27, P < 0.001), and the Margalef’s index contributed to 75% of the variation (Fig. 3).

Figure 2 Relationships between C:N:P stoichiometric ratio and (A, D, G) bulk density, (B, E, H) fine root biomass and (C, F, I) soil moisture at 0–30 cm of soil profiles in 2018.

Discussion

Soil C:N:P ecological stoichiometry for plantations in subalpine region

In our experiment soil C:N at the depth of 0–30 cm ranges from 14.5 to 15.5 in the examined ecosystems, which is slightly higher than the global average C:N of 14.3 (Yue et al., 2017) and lower than the other subalpine average C:N (Bing et al., 2016). Soil C:P at the depth of 0–30 cm ranges from 184 to 299 in the examined ecosystems, which is higher than China’s average of 136 (Tian et al., 2010), lower than the global average and the other subalpine (Bing et al., 2016; Yue et al., 2017). Soil N:P at the depth of 0–30 cm ranges from 12.9 to 19.4, which is higher than that of global and China’s average (9.3 and13.1, respectively) (Tian et al., 2010; Yue et al., 2017) and lower than the other subalpine average C:N (Bing et al., 2016). What account for the discrepancy across studies are largely unknown.

Table 3 Correlations among soil C:N:P stoichiometric ratios and plant community attributes in 2018.

Variables	C:N	N:P	C:P	
Tree biomass	−0.002NS	−0.887**	−0.833**	
Understory plant biomass	−0.348NS	0.894**	0.516NS	
Litter stock	0.862**	−0.212NS	0.278NS	
Total aboveground biomass	−0.006NS	−0.875**	−0.825**	
Tree biomass/Total aboveground biomass	0.371NS	−0.954**	−0.680*	
Understory plant biomass/Total aboveground biomass	−0.502NS	0.910**	0.563NS	
Fine root biomass	-.632*	0.292NS	−0.044NS	
Litter carbon	−0.595*	−0.279NS	−0.591*	
Litter nitrogen	0.149NS	0.271NS	0.356NS	
Litter phosphorus	0.03NS	−0.587*	−0.535NS	
Litter C:N	−0.339NS	−0.434NS	−0.610*	
Litter N:P	0.063NS	0.546NS	0.561NS	
Litter C:P	−0.455NS	0.423NS	0.143NS	
Richness index	−0.129NS	0.967**	0.833**	
Margalef index	0.054NS	0.902**	0.878**	
Shannon-Wiever index	0.193NS	0.851**	0.906**	
Simpson index	−0.35NS	−0.730**	−0.876**	
Pielou index	0.42NS	0.670*	0.860**	
Notes.

NS Not significant

* P < 0.05

** P < 0.001

Soil C:N:P ecological stoichiometry between tree species

In accordance with our first hypothesis, soil C:N:P stoichiometry varied significantly with tree species (Table 2; Fig. 1), particularly for the topsoil, where C:N ratios in LAR and PT plantations are greater than CJ plantation and shrubland, implying higher N mineralization rate in shrubland. Three likely reasons account for this finding. Firstly, litter inputs and stocks differ across the examined plantations (Table 4). In the topsoil, LS was the most influencing factor of soil C:N (Fig. 3). Secondly, the microclimate, the quantity and quality of root exudates and rhizodeposits as well as soil microbial community change with plant species (Aoki, Fujii & Kitayama, 2012; Ohta & Hiura, 2016; Zhang et al., 2011), which may jointly influence soil nutrient status and its stoichiometric ratio. Firstly, as shown by previous studies, broadleaf litter is more decomposable than needle litter in boreal forests (Laganiere, Pare & Bradley, 2010). Besides, allocation of C to roots is directly proportional to photosynthesis (Brzostek et al., 2015) and understory shrubs generally have a lower photosynthetic capacity than overstory trees (Sakai et al., 2005). Nevertheless, the C:N ratio of conifer stands is greater than broadleaf stands may be related to the canopy density and high light interception of conifers reduce the light efficiency on the forest floor (Lieffers et al., 1999). Furthermore, the decomposition rates of broadleaf trees are commonly higher in comparison with conifer trees (Taylor, Parkinson & Parsons, 1989; Zhang et al., 2019).

Figure 3 The gradient boost decision tree measuring the relative importance of factor influencing topsoil (0–10 cm) C:N:P stoichiometric ratio in 2018.

(A) C:N, (B) N:P, (C) C:P. FRB, Fine root biomass; LS, Litter stock; TB:TAB, Tree biomass/Total aboveground biomass; MI, Margalef index; L-P, Litter phosphorus; L-C, Litter carbon. *P < 0.05, **P < 0.001.

Table 4 The different plantations component of biomass in this study area.

Species identity	TB (t⋅ha−1)	UPB (t⋅ha−1)	LS (t⋅ha−1)	TAB (t⋅ha−1)	TB:TAB	UPB:TAB	
LAR	91.63 ± 13.35bc	0.29 ± 0.095b	8.35 ± 0.41a	100.27 ± 13.68bc	0.91 ± 0.010c	0.005 ± 0.0004b	
PT	157.16 ± 37.57ab	0.26 ± 0.037b	7.72 ± 0.44a	165.13 ± 38.03ab	0.95 ± 0.012b	0.003 ± 0.0009b	
CJ	210.65 ± 8.20a	0.25 ± 0.147b	3.94 ± 0.17b	214.84 ± 8.29a	0.98 ± 0.001a	0.002 ± 0.0006b	
S	29.26 ± 2.53c	10.69 ± 1.308a	3.87 ± 0.89b	43.81 ± 3.41c	0.67 ± 0.012d	0.075 ± 0.0061a	
Notes.

TB Tree biomass

UPB Understory plant biomass

LS Litter stock

TAB Total aboveground biomass

TB:TAB Tree biomass/Total aboveground biomass

UPB:TAB Understory plants biomass/Total aboveground biomass

Lowercase letters indicate significant differences between tree species (P < 0.05).

Vertical pattern of soil C:N:P ecological stoichiometry

Consistent with our second hypothesis, the C:N, C:P and N:P of soil decreased with increase in the soil depth (Fig. 1). This finding is in agreement with previous studies addressing vertical pattern of soil C:N:P stoichiometry in forest soils (Feng, Bao & Pang, 2017; Li et al., 2013; Qiao et al., 2020; Tian et al., 2010; Tischer, Potthast & Hamer, 2014). These results are maybe related to the fact that soil nutrients decreased with soil depth. Besides, this could be due to the topsoil layer environmental factors being easily affected and the return of nutrients from litters (Feng, Bao & Pang, 2017). In addition, soil nutrients are usually first concentrated on the topsoil and then transferred to the subsoil layer with water or soil animals. Furthermore, soil C:N ratio decreased with the soil depth among different plantations, and this could be related to the different nutrient turnover rates in decomposition process. The easily decomposable material elapsed and N is fixed in decayed products and microbial biomass, remaining durable materials had more slowly decomposition rates and lower C:N ratio (Yang et al., 2010). Compared with the topsoil layer, the organic matter in subsoil layer is more humified and older, result in continuous decrease of the soil C:N ratio with soil depth (Callesen et al., 2007; Yang et al., 2010). Additionally, difference in soil nutrient associated with changes in soil microbial dynamics, litter decomposition, microbial food web, and soil nutrient accumulation and circulation (Griffiths, Spilles & Bonkowski, 2012; Zhao et al., 2015). Besides, the decrease in soil temperature with the increase of soil depth (Jackson et al., 2000) may account for the decreased soil C:N:P stoichiometric ratios in lower depth.

Potential influencing factors of soil C:N:P stoichiometry

In partial agreement with our third hypothesis, linkage among soil C:N:P ecological stoichiometry, soil properties and plant community attributes varied with soil depth. This is also in agreement with our earlier result (Feng, Bao & Pang, 2017), the relative contribution of factors varied among soil depths and the examined object. Firstly, bulk density increased with the increase in soil depth, result in decreased soil porosity, which further reduce soil moisture and root penetration. Besides, the topsoil of trees is in direct contact with the plant community. Hence, LS, TB/TAB and other plant community structure index will effect on it. Additionally, the correlations between environmental factors and stoichiometric ratios depended on the elements considered. In summary, the effects of tree species and soil depth on soil C:N:P stoichiometry associated with bulk density, soil moisture, the quantity and quality of aboveground litter inputs and underground fine root.

Conclusions

We observed that tree-specific and depth-dependent have strong effects on soil C:N:P stoichiometry in subalpine plantations. In general, the C:N, C:P and N:P of topsoil are higher in comparing with that of subsoil layer at 0–30 cm depth profiles. The difference in C:N, N:P and C:P at the topsoil across target tree species significantly linked to standing litter stock, tree biomass/aboveground biomass and Margalef’s index of plant community, respectively, whereas the observed variations of C:N, N:P and C:P ratio among soil profiles are closely related to differences in soil bulk density, soil moisture, the quantity and quality of aboveground litter inputs and underground fine root across plantations examined. Our results highlight that soil nutrients status in plantation depend on litter quantity and quality of selected tree species, as well as soil physical attributes. Therefore, matching site with trees is crucial to enhance ecological functioning in degraded regions resulting from human activity.

Supplemental Information

Supplemental Information 1 Original data

1. Basic data; 2. related data of C:N:P

Click here for additional data file.

We thank De feng Feng and Xin Liu for helpful suggestions on the study and earlier version of the manuscript. We also highly appreciated Long Huang, Shuang ping Peng and Zhong ping Tang for their assistance and support provided in sample collection and analysis.

Additional Information and Declarations

Competing Interests

Author Contributions

Data Availability

The authors declare there are no competing interests.

Kai bin Qi conceived and designed the experiments, performed the experiments, analyzed the data, prepared figures and/or tables, authored or reviewed drafts of the paper, and approved the final draft.

Xueyong Pang performed the experiments, analyzed the data, authored or reviewed drafts of the paper, and approved the final draft.

Bing Yang analyzed the data, authored or reviewed drafts of the paper, and approved the final draft.

Weikai Bao conceived and designed the experiments, performed the experiments, analyzed the data, authored or reviewed drafts of the paper, and approved the final draft.

The following information was supplied regarding data availability:

Raw data are available in the Supplemental File.

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
