# Peer review of "Soil carbon, nitrogen and phosphorus ecological stoichiometry shifts with tree species in subalpine plantations"

_PeerJ, doi:10.7717/peerj.9702_

## Round 0.1 · original submission · Major Revisions

Dear Dr Bao and co-authors,

I have received two reviews from discipline experts. Please pay attention to all of their comments, suggestions and guidance with a detailed response letter and track changes to a revised manuscript.

Please pay particular attention to:
1) The Abstract requires rewriting and better indication of major findings that relate to the research objective
2) All references should be carefully checked and brought up to date where appropriate.
3) Please improve the description of the experimental design and statistical analysis
4)The results can be shorter in description of only the major findings of relevance to the research questions.
Please get a colleague proficient in English scientific writing to assist, or pay for English editing assistance for the manuscript overall.

We look forward to receiving a thoroughly revised manuscript.
Regards
Steve

Reviewer 1 ·

Basic reporting

see "General comments for the author".

Experimental design

see "General comments for the author".

Validity of the findings

see "General comments for the author".

Additional comments

The study of “Effects of reforestation on soil carbon, nitrogen, phosphorus stoichiometric shifts depend on tree species in subalpine forests” by Qi et al addressed soil C:N:P ratio in different soil depth with various tree species. This study is meaningful in the reforestation field. However, many points should be attentioned in this study. Firstly, many results were described simply, especially for the section of Abstract and Results. Secondly, I suggest rewrite the section of Abstract, the present writing is not good. Thirdly, references should be carefully checked, many mistakes. Moreover, I suggest that increase some references of recent years. Furthermore, I suggest theauthors find a native English speaker to improve language. Therefore, I suggest major revision for this study.

Many points are mentioned here, but not all, as follows:

Abstract
L14, address the important meaning of studying soil stoichiometric ratio.
L18, list them in the first appearance.
L19-23, specific results should be described in this section. Describing how it changed, how much did it change, need more numbers to describe it.
L28, In conclusion…. I do not suggest the conclusion. I suggest that you should highlight the theme of this study.

Introduction
L34, I suggest “Nitrogen (N) and phosphorous (P) are the most important nutrients promoting plant growth,”. Moreover, you should also say some points to C.
L36, delete point.
L37, define C.
L38-41, grammar mistake. I can not understand the sentence.
L43, delete in an ecosystem.
I suggest the second paragraph can be put as the first paragraph, and the first paragraph can mix together with the second paragraph.
L72-73, I suggest it is not a good hypothesis.

Results
L149-184, you should detailedly describe the results. For example, “the soil C was X.X time higher in 0-10 cm than 20-30 cm”. Describing the results by using some numbers.

Discussion
L212-220 I suggest many of them can put into the section of Introduction.
L221-226 Many expression are too absolute. You can change some expression.
L228-229 Grammar mistake.
L247 results showed.
L251-254 grammar mistake.
L258-259 reference mistake.
L263 our change with the

References
All the references should be rearranged.
Lacking of year of reference. For example, L299, L311, L331, L339-345, L351, L364, L371, L373, L379, L381, L399.
Some references list all authors, while some references list three authors.
Moreover, there were more small mistakes in this section, you should carefully check.

Table 3, AL-biomass: I suggest S is “c”, not “a”. Please check.
Fig. 3 increase standard error.

Reviewer 2 ·

Basic reporting

The English standard is adequate but would be improved by the services of a professional editor. I have made some minor comments to improve English throughout but have not made this my aim.
The references are sufficient, formatting of citations and corrections in the reference list are needed.
Structure is adequate and raw data provided - however there needs to additional clarification in the raw data provided.

Additional information is provided in my general comments to the author.

Experimental design

Hypotheses are posed and meaningful, however declines in soil nutrients with increasing soil depth are well-established so that findings in this area are not novel - a greater focus needs to be attention on how tree species moderate the soil depth effects.
The experimental design needs greater description and statistical analysis and presentation needs attention.

Additional information is provided in my general comments to the author.

Validity of the findings

Conclusions are well stated, but these findings are lost in the excessive description of results - the focus needs to be on the major findings rather than listing all significant results. The abstract would be improved by translating major findings described in the conclusion.
Some of the findings may need to be revised depending on the detail of the experimental design (that is lacking) - with possible implications for pseudo-replication.

Additional information is provided in my general comments to the author.

Additional comments

ABSTRACT
The abstract does not seem to specifically address each of the three hypotheses posed in the introduction.
L19, remove ‘greatly’ (non-descript)

INTRODUCTION
L38-41, The first sentence seems incomplete (missing ‘regulate’ perhaps) and the second comes across as a statement of fact without adding to the message of the first paragraph. For example, what are the consequences of N and P deficiencies (the fact that C:N, C:P ratios increase is self-evident)? These two sentences could follow the first sentence in the paragraph, with consequences on N, P deficiencies/excess to follow. This would improve on the current disjointed nature of the paragraph.
L73, what are ‘better’ soil nutrient conditions?
L75, dos this mean that climate, topography, elevation are consistent across the different plantations examined?
MATERIALS AND METHODS
L98, reference for evidence of this?
L103-104, more detail is required for the sampling strategy and in particular whether there were multiple plantations for each species, or whether the three plots were located within a single plantation of each species. Plantation size, the relative location of plots from each other and the relative location of plantations to each other are all important. These each have a role to play on possible spatial autocorrelation – which is ideally considered either as part of your experimental design, or within the statistical analyses. There may be different underlying factors within each plantation type that are dictating your results above and beyond that of species effects. If sampling is limited to one plantation of each species then the sampling approach could be considered pseudo-replication with flow on effects to the applicability of results (i.e. limited to a ‘case study’). Please provide a more detailed description of your sampling design and possible implications (e.g. in relation to pseudo-replication).
Given one hypotheses relates to broadleaf species, you should make clear to the international audience which of the examined species were broadleaved.
L106, what are the layers of understorey vegetation (sampling categories) that were sampled?
L114, what do you mean by treatment? It’s not clear how many bulk density samples were retained and analysed for each 10 x 10 m plot? It’s also not clear to what temperature the soil bulk density samples were dried (air dried rather than dried to constant weight?)
L115, was soil bulk density corrected for gravel/stone content, if so how was volume estimated?
L124-126, please confirm that contents were used as the raw data includes results for concentrations.
L136, is this a valid approach given significant interaction effects between tree species and soil depth (i.e. more appropriate to examine for example, tree species effects within depth)?
L138, what are the plant community indices used?
RESULTS
L150, avoid descriptors such as ‘greatly’
L153, you mean ‘responsive’?
L150-155, this could be shortened since they each had significant tree / depth / interaction effects.
L157, ‘of’, not ‘on’
L157-184, rather than detailing each significant effect and a laundry list of results, aim to communicate the main message(s), e.g. whether stoichiometry in plantations differed to that in shrubland, whether this differed with time and soil depth. The level of detail currently provided means that the message gets lost.
L188, are these results across both years? Is this regression rather than correlation? Please provide a table of results for the correlation coefficients (to indicate direction of relationship also – and results for variables not provided). In the methods you need to provide some detail on what components of biodiversity were used to calculate diversity indices.
L188-210, similar to earlier comments, this section needs to be shortened to focus on the main results.
DISCUSSION
L212-215, repetitive, please revise
L212-220, this is introductory material that would be useful to include in your introduction to strengthen arguments around the utility of stochiometric ratios. Better to begin your discussion with a statement of the most significant findings.
L221-226, this is descriptive information and while interesting, how does it address the three hypotheses posed in the introduction? Ideally you should structure your discussion to clearly address the hypotheses posed in the introduction [this is done further down in the discussion – better to get ‘straight to the point’].
L236, can you add to discussion about differences in litter quality, e.g. > lignin content in conifers relative to broadleaved species.
L250, ‘what other medium’, what about the role of decomposers in vertical redistribution of soil nutrients?
L267, was soil bulk density measured through time (data file only shows one year of collection).
CONCLUSION
Draws out the main messages well – these messages need to be better incorporated into the abstract.
CITATIONS
These are inconsistent throughout and need to be reviewed for consistency.
TABLE 1
‘C-density’ is described in the header but not used as a column identifier; revise. Please place the plantation abbreviation in the first row of the associated data – at present (i.e. middle justified) it is difficult to tell where one plantation starts and ends. Ensure also consistency in decimal places within units.
TABLE 2
‘species’ not ‘spices’
FIGURE 1
The use of superscripts in the figures (capital, lower case letters) needs to be corrected. The use of upper and lower case is only appropriate where there are significant interaction effects (i.e. tree species x time). Where there are no interaction effects, tree species effects need to be across both years (as effects are consistent) – for example for 0-10 cm, letters should be consistent within years for each plantation as there were no interaction effects.
FIGURE 2
Is this across both years? The raw data doesn’t provide any bulk density (for example) data for 2007.
RAW DATA
For the basic information, please provide plantation identifiers to match those in the ‘CNP’ tab.
For 2007, data is limited to ratios, please provide the complete data set (i.e. to match 2018).

---

## Round 0.2 · Minor Revisions

Dear Dr Bao and co-authors,

I have received further reviews from the two expert academics that previsouly reviewed your manuscript. I thank them for their rapid attention and continued interest in improving your manuscript.

Both reviewers suggested minor revisions were required, and I am in accordance and recommend that the manuscript requires further minor but necessary revisions to be ready for publication.

Both reviewers recognise that there continue to be deficiencies in the English grammar used, please regard this an important issue to resolve.

R2 points out that there continues to be a mismatch between the statistics and results. Please address this.

Several tables and figures require attention and some require citation in the manuscript text to be relevant or necessary.

Both reviewers provide some detailed comments to assist you in rapidly addressing these necessary further revisions. I look forward to receiving a second revision of your manuscript in the coming weeks.

Regards
Steve

Reviewer 1 ·

Basic reporting

no comment

Experimental design

no comment

Validity of the findings

no comment

Additional comments

I have read the paper again, significant improvements have been found in the R version. I only have few points here.

1. L138, increase full name of SOC and TN.
2. L200, the title is too long to understand.
3. References: increase some new papers, such as 2019-2020.
4. The English expression of the title and note in Figures and Tables should be carefully checked, many grammar mistakes.

Reviewer 2 ·

Basic reporting

Manuscript could be further improved by attention to grammar. Table S1, S3 not referred to in text. Corrections needed in Figs and Tables.

Experimental design

Further detailed required in relation to spatial locations of plantations - see comments to authors.

Validity of the findings

There is a mismatch between the statistics and results - this needs to be addressed.

Additional comments

GENERAL COMMENTS
There are a number of outstanding items in relation to presentation of results and discussion material that still require attention. Detailed comments provided that I hope the authors find useful.
TITLE
Revise: Soil carbon, nitrogen and phosphorus ecological …..
ABSTRACT
L23, plantations NOT plantation
L24, ecologically NOT ecological
L25, remove space before comma
L27, ‘and’ before Cercidiphyllum
L28, ‘a’ before ‘subalpine’
I will refrain from further English corrections – there is still some work to be done in this regard.
L29, not clear what topsoil and subsoil is (the topsoil could be 0-30 cm, and subsoil greater than 30 cm, clarify)
L31, what is ‘standing litter stock’? Litter is not standing biomass?
KEY WORDS
Revise to key words rather than phrases, suggest ‘tree species’ is not useful
INTRODUCTION
L59, regarded NOT regard
L60, nutrients NOT nutrient
L67, this is not a sentence
L75, this is not a sentence
L77, remove ‘that’; C:N:P NOT C/N/P
L78, It is not surprising that results vary among studies – this is true for all studies of soil forest nutrition, soils, plant-soil feedbacks – you need a greater argument for the significance of your study than results vary across studies.
L81, but your study does not examine ‘spatial pattern’?
L86, with NOT will
There is no mention of ‘restoration’ in the introduction or discussion – this creates a disjoint between the study purpose, findings and the all important abstract – please address.
MATERIALS AND METHODS
L104-117, there are still some outstanding points in relation to the initial review that need to be addressed, specifically, “Plantation size, the relative location of plots from each other and the relative location of plantations to each other are all important. These each have a role to play on possible spatial autocorrelation – which is ideally considered either as part of your experimental design, or within the statistical analyses. There may be different underlying factors within each plantation type that are dictating your results above and beyond that of species effects.”
I suggest you provide detail on the size of the areas sampled and include a figure in the appendix to indicate relative location of plantations / sites from each other.
L169, 0-30 cm NOT 0-30cm, maximum NOT maximal
RESULTS
Previous comments in relation to avoiding the use of descriptors such as ‘greatly’ are still outstanding, e.g. L175 (and address throughout).
L176, ‘responsive responsible’?
L179-187, the description of the results are not consistent with the statistics. The only significant interaction between species and time is for 10-20 cm depth, however time x tree species differences are also referred to for 0-10 cm depth despite no interaction? There is no reference to 20-30 cm results for C:N.
L188-191, what about significant species effects for 20-30 cm?
L192-199, but results for 0-10 cm are consistent across time?
L200-202, this is too long for a sub-heading
L208, by ‘arbor’ do you mean ‘tree’ – arbor can have a number of meanings
L218, 221, 223, remove decimal places
DISCUSSION
Given the discussion presents hypothesis 1, 2 and 3, these should be clearly indicated as such in the introduction.
L226-231, how do results compare with other subalpine forests?
L245, litter quality, and greater lignin content of conifer litter relative to broadleaf species is critical here – expand.
L249-266, there is no discussion of depth effects in the results section – address.
L249, Fig 1 doesn’t not show significance of depth effects, Table 2 most relevant here.
L254, not clear what you mean by ‘more sensitive’?
L256, ‘other medium, meaning?
L268-279, this Is largely results material and repetitive of the results – more discussion needed here.
TABLES/FIGURES
Table 1, what does ‘canopy’ refer to
Figure 1, the use of letters to indicate significance of differences is still not correct. For example in Fig 1D, tree species differences are consistent across time and with no species x time interaction – this is not what the letters indicate. Similarly Fig 1A, tree species differences are consistent across time, but the letters suggest tree species differences in 2018, but none for 2007. Address all panels and correct. Fig 1G also incorrect.
Figure 2, insert space before (0-30cm) and space before ‘cm’, similar for Figure 3
Table S1 and S3 are not referred to

---

## Round 0.3 · Minor Revisions

Dear Dr Bao and co-authors,

Thank you for your diligent and detailed attention to the reviewer's comments and suggestions.

As Editor handling this manuscript I can still see several issues that I would request that you resolve.

Firstly, R2 clearly requested that you make a clear link from the Introduction to the Discussion with regards to the three hypotheses being tested:

“DISCUSSION Given the discussion presents hypothesis 1, 2 and 3, these should be clearly indicated as such in the introduction.”
You have responded to this satisfactorily. To make it very clear for I am requesting that you use the word(s) "hypothesis" in the Introduction and you use and indication of 1, 2 and 3 (or first, second and third as in the Discussion) so that the reader can more easily see a hypothesis bridge from the Introduction to the Discussion. This is important and will make the manuscript far more logical and appealing for the reader.

Secondly, The additional text in the Introduction requires attention. I am adding this text here and indicating the changes I suggest:

"In addition, ecological stoichiometry can also be influenced by ecological restoration approaches, and afforestation can be a major factor affecting soil C:N:P stoichiometric ratio (e.g., Davis et al. 2007; Deng et al. 2014). Different tree species planted for restoration of afforestation can influence soil properties through multiple pathways. However, in alpine and subalpine regions whether there are differences in soil C:N:P stoichiometric ratio between afforestation and reforestation remains largely unknown. Moreover, tree species differ from one another in the quality and quantity of litter and root exudate (Aponte et al. 2012; Paterson et al. 2007) and this in turn can influence soil C and N mineralization mediated by microbial communities (Alberti et al. 2015; Prescott & Grayston 2013)."

Once you are able to attend to these two important changes, please resubmit for final consideration.

Regards
Steve

---

## Round 0.4 · accepted · Accept

Congratulations on a job well done - thank your patience and diligence in improving this manuscript which is now fit for publication.

Regards
Steve